# Is Elo Rating Reliable? A Study Under Model Misspecification

## Abstract

Elo rating, widely used for skill assessment across diverse domains ranging from competitive games to large language models, is often understood as an incremental update algorithm for estimating a stationary Bradley-Terry (BT) model. However, our empirical analysis of practical matching datasets reveals two surprising findings: (1) Most games deviate significantly from the assumptions of the BT model and stationarity, raising questions on the reliability of Elo. (2) Despite these deviations, Elo frequently outperforms more complex rating systems, such as mElo and pairwise models, which are specifically designed to account for non-BT components in the data, particularly in terms of win rate prediction. This paper explains this unexpected phenomenon through three key perspectives: (a) We reinterpret Elo as an instance of online gradient descent, which provides no-regret guarantees even in misspecified and non-stationary settings. (b) Through extensive synthetic experiments on data generated from transitive but non-BT models, such as strongly or weakly stochastic transitive models, we show that the "sparsity" of practical matching data is a critical factor behind Elo's superior performance in prediction compared to more complex rating systems. (c) We observe a strong correlation between Elo's predictive accuracy and its ranking performance, further supporting its effectiveness in ranking.

## 1 Introduction

The Elo rating system, introduced by Arpad Elo (Elo, 1961), is a widely-used method for rating player strength in two-player, zero-sum games. Initially developed for chess, Elo has since been adopted across a broad range of games, including Go, sports, video games, and recently, in evaluating large language models (LLMs) and AI agents. Elo rating is usually interpreted as an incremental update algorithm for estimating an underlying stationary Bradley-Terry (BT) model. BT model assumes each player $i$ has a scalar strength rating $\theta[i]$ (which does not change), and for a single game between player $i$ and $j$, the probability that player $i$ wins is $\sigma(\theta[i] - \theta[j])$, where $\sigma$ is the logistic function. Based on this model, after each game, Elo rating system will adjust each player's rating according to the actual game result.

Despite the widespread use of Elo, its foundation on games following stationary BT models appears restrictive. In this paper, we first examine whether the BT assumption holds in real-world datasets. Using a likelihood ratio test, we show that game outcomes in many datasets deviate significantly from the BT model, indicating substantial model misspecification. Furthermore, we observe that player skills and matchmaking distributions are often non-stationary. This raises serious concerns over Elo's reliability on practical uses. Surprisingly, we also observe that, despite these deviations, Elo still frequently outperform more complex models, such as mElo and pairwise methods—designed to capture non-BT components—in predicting outcomes of the real-world games. These findings call for a deeper understanding of Elo beyond its conventional interpretation as a BT model parameter estimator. In this paper, we explore this phenomenon through three key perspectives.

First, we interpret the Elo rating system through the lens of regret minimization. Specifically, Elo can be seen as an instance of online gradient descent—an online convex optimization (OCO) algorithm with sublinear regret guarantees, even in adversarial settings. This covers both non-stationary environments and data that deviate from the BT model. Consequently, Elo performs well as long as

the best BT model in hindsight provides a reasonable fit to the data and sufficient data is available to keep regret small.

Second, we conduct synthetic experiments to systematically evaluate different algorithms in controlled settings. We test on transitive but non-BT games, including those following strongly and weakly stochastic transitive models, and introduce non-stationary factors such as Elo-based matchmaking and dynamic player strengths to better reflect real-world scenarios. Our findings reveal that data "sparsity" plays a crucial role in prediction performance, driven by a trade-off between model misspecification error and regret. In sparse datasets—where the number of games per player is low—regret becomes the dominant factor in performance, favoring simpler models like Elo, which incur lower regret despite higher misspecification error. In contrast, more complex models such as mElo and pairwise methods achieve lower misspecification error but suffer from a significantly higher regret. We confirm that many real-world games operate in this "sparse" regime, explaining Elo's strong empirical performance. However, in "dense" regimes, where players engage in more games, Elo is outperformed by more complex models when applied to non-BT data.

Finally, we also investigate the ranking performance of Elo. For pairwise ranking, we find a strong correlation between prediction accuracy and ranking accuracy. However, we caution that Elo should not be blindly trusted, as it can fail to produce consistent total orderings under arbitrary matchmaking, even in transitive datasets.

In summary, our contributions include: (1) Demonstrate that real-world game data often violates the BT model via likelihood ratio tests. (2) Show that Elo achieves strong predictive performance even in non-BT datasets. (3) Interpret Elo through a regret-minimization framework, proving its effectiveness in nonstationary setting under model misspecification. (4) Highlight the role of data sparsity in algorithms' prediction performance, with extensive synthetic and real-world experiments. (5) Explore the correlation between prediction accuracy and ranking, theoretically study Elo's ranking performance under different matchmaking setups.

## 1.1 Related work

**Methods for rating game players**  A large number of rating methods used in practice can be viewed as variants of Elo, most notably Glicko (Glickman, 1995), Glicko2 (Glickman, 2012) and TrueSkill (Herbrich et al., 2006). A common characteristic shared by these methods is that they assume a scalar rating for players with parametric probabilistic model (Bradley-Terry in Glicko and Thurstone in TrueSkill) and make incremental gradient-like updates for each game or a small batch of games. mElo (Balduzzi et al., 2018) and Disk Decomposition (Bertrand et al., 2023) generalize Elo score by rating every player with a multi-dimensional vector instead of a scalar. Their approach can be understood as low-rank approximation of the logits of the winning probabilities. In our work we regard them as Elo2k, and examine their performance is a central part of our work.

Bayeselo (Coulom, 2005) and WHR (Coulom, 2008) are two popular Bayesian methods that are also based on the BT model. They differ from Elo-like incremental updates by requiring more compute to produce a maximum a posteriori estimator every step.

**Analysis of Elo score**  Despite its popularity and wide applicability, the analysis of Elo score is "curiously absent" (Aldous, 2017). Elo discussed practical concerns and small-scale statistical validations of the method in Elo (1978). Most related to this work, however, is the proposal of the linear approximation of the update formula. Aldous (2017) proved the existence and uniqueness of a stationary distribution under the Elo update rules without assuming realizability. However, the nature of this stationary distribution is not explored. de Pinho Zanco et al. (2024) analyzed the convergence of Elo score assuming round-robin match making, realizability of the Bradley-Terry model and linearization of $\sigma$. For more empirical and simulation results, see Király & Qian (2017) and references within.

**Misspecification of the Bradley-Terry Model**  The Bradley-Terry (BT) model and related parametric preference models have been criticized for being inaccurate models of human preferences (Ballinger & Wilcox, 1997). For example, Oliveira et al. (2018) show that the BT model does not fit match data from ∼200 computer chess programs.Bertrand et al. (2023) further demonstrate that extending BT to a $k$-dimensional model improves prediction on synthetic datasets (Czarnecki et al.,

2020),suggesting misspecification. However, their synthetic datasets are payoff matrices (between each pair of players), which differ significantly from real game datasets which are typically sparse and binary.

A line of work focus on testing the independence of irrelevant alternatives (IIA) assumption (Hausman & McFadden, 1984; Seshadri & Ugander, 2019; Cheng & Long, 2007), which implies the BT model in pairwise comparisons. However, rejecting IIA may not suffice to reject BT, since it is a weaker condition. More recently, Makur & Singh (2023) developed a minimax test for BT, but it assumes each pair is compared multiple times, making it impractical for sparse datasets like Chess or Go, where many player pairs never compete. In contrast, our proposed test applies even in such sparse regimes.

Finally, the BT model assumes transitivity, yet non-transitive or cyclic behavior is well-documented in games (Samothrakis et al., 2012; Chen & Joachims, 2016; Omidshafiei et al., 2019; Czarnecki et al., 2020). While detecting such violations would refute BT, statistical testing is difficult when most player pairs have no match data.

**Learning to rank**    There has been a long line of work studying various flavors of learning-to-rank (for instance, see Liu et al. (2009); Negahban et al. (2012); Braverman & Mossel (2009); Shah & Wainwright (2018) and references within), where the focus is to construct a global ranking based on a dataset partial observations. While highly relevant to task of rating game players, we note that these methods generally receive less attention in game-related applications. These methods are typically not able to predict win-loss probability of a particular matchup either. For these reasons, we focus on understanding Elo and the rating systems within the family of Elo in the scope of this work. We left the connection and comparison to other learning-to-rank methods as important future directions.

## 2 PRELIMINARY

We consider the scenario where $N$ players play against each other in a sequential manner. Specifically, for every $t \in [T]$, players $i_t \in [N]$ and $j_t \in [N]$ are chosen by the matchmaking scheme to play against each other at time $t$. The outcome $o_t \in [0, 1]$ denotes the utility of player $i_t$, which can be chosen as 1, $1/2$ and 0 to denote a victory, a draw and a loss respectively; Player $j_t$ receives utility $1 - o_t$. There are two main tasks in this setting. The first task is **prediction**, i.e., predicting the outcomes of the game. At time $t$, the learner is tasked to gives a prediction $p_t$ for the player $i_t$'s win rate against $j_t$, after observing the previous games $\{(i_k, j_k, o_k)\}_{k=1}^{t-1}$ and the two players at the current round $(i_t, j_t)$. It is natural to evaluate the prediction accuracy of the algorithms by binary cross entropy loss

$$\ell_t := -(o_t \ln p_t + (1 - o_t) \ln(1 - p_t)). \tag{1}$$

The accumulated loss until time $t$ is $\mathcal{L}_t := \sum_{i=1}^{t} \ell_i$. The second task is **ranking**, i.e., give a total order or pairwise order for all players according to their relative strength. A total order ranking is well-defined only if the underlying game has a transitive structure. For simplicity of discussion, this paper will mostly focus on prediction, and leave the discussion of ranking to Section 4.3.

### 2.1 ALGORITHMS

Here, we introduce several important and representative online rating algorithms.

**Elo rating:**    Elo maintains a *scalar* rating (which is also refered as *score*) for each player. Concretely, let $\theta_t \in \mathbb{R}^N$ denote the ratings of all players at time $t$, then Elo scores can be computed using updates:

$$\begin{cases} p_t & \leftarrow \sigma\left(\theta_t[i_t] - \theta_t[j_t]\right), \\ \theta_{t+1}[i_t] & \leftarrow \theta_t[i_t] + \eta_t\left(o_t - p_t\right), \\ \theta_{t+1}[j_t] & \leftarrow \theta_t[j_t] - \eta_t\left(o_t - p_t\right). \end{cases} \tag{2}$$

Here $\sigma = 1/(1 + e^{-x})$ is the logistic function. Elo is often understood under the assumption that the outcome of the game is sampled from the Bradley-Terry (BT) model:

$$\mathbb{P}[o_t = 1 | i_t, j_t] = \sigma(\theta^\star[i_t] - \theta^\star[j_t]). \tag{Bradley-Terry}$$

where $\theta^\star[i]$ represents the *true score* of player $i$. In this case, Elo update is simply an incremental update algorithm for estimating the parameters $\theta^\star$ of the BT model.

| Dataset | $N$ | $2T/N$ | Test Statistic | $p$-value | Accuracy of Rating Algorithms | | | | |
|---|---|---|---|---|---|---|---|---|---|
| | | | | | Elo | Glicko | TrueSkill | Elo2k | Pairwise |
| Renju | 5k | 49.8 | 150.0 | $< 10^{-10}$ | 0.6039 | 0.6100 | 0.5995 | 0.6109 | 0.6688 |
| Chess | 185k | 125.4 | 2020.1 | $< 10^{-10}$ | 0.6391 | 0.6349 | 0.6308 | 0.6387 | - |
| Tennis | 7k | 52.5 | 37.3 | $< 10^{-4}$ | 0.6242 | 0.6232 | 0.6209 | 0.6365 | 0.6820 |
| Scrabble | 15k | 200.7 | 142.2 | $< 10^{-10}$ | 0.6730 | 0.6766 | 0.6756 | 0.6787 | 0.6894 |
| StarCraft | 22k | 38.7 | 775.8 | $< 10^{-10}$ | 0.5713 | 0.5689 | 0.5828 | 0.5832 | 0.6753 |
| Go | 426k | 60.4 | 193411.2 | $< 10^{-10}$ | 0.6443 | 0.6375 | 0.6321 | 0.6372 | - |
| LLM Arena | 129 | 23156.9 | 73.1 | $1 \times 10^{-3}$ | 0.6607 | 0.6602 | 0.6611 | 0.6611 | 0.6619 |
| Hearthstone | 27 | 4626.1 | 49.0 | $< 10^{-4}$ | 0.6898 | 0.6893 | 0.6894 | 0.6847 | 0.6853 |

Table 1: Summary of real-world dataset experiments. "Test Statistic" and "p-value" indicates the test statistic and the corresponding p-value; "Accuracy of Rating Algorithms" indicates the cross entropy loss at time $T$ of each algorithm.

**Glicko, TrueSkill — "Elo-like" rating:**    The second class of rating systems we examine consists of Glicko (Glickman, 1995) and TrueSkill (Dangauthier et al., 2007). Similar to Elo, they assume total ordering among players, and mainly use a scalar rating to represent the relative strength of each player. Different from Elo, Glicko additionally introduces a "rating deviation" parameter for each player which measures the uncertainty in the rating. Trueskill is similar to Glicko, but instead assuming the outcomes are sampled from a different probabilistic model, which changes the $\sigma$ function in BT models from logistic function to the cumulative distribution function of Gaussian, up to proper renormalization.

**Elo2k, Pairwise — more complexity rating systems:**    These systems are much more flexible than Elo, Glicko, TrueSkill—they no longer assume the total order among players, and are able to model cyclic structure among players (i.e., player A beats player B, player B beats player C, and player C beats player A). In particular, Elo2k generalizes Elo by assign each player with a vector rating of dimension $k$, instead of a scalar rating. It is also known as mElo (Balduzzi et al., 2018) or Disk Decomposition (Bertrand et al., 2023). This algorithm has $Nk$ parameters. **Pairwise** simply computes the pairwise win rate for each pair of players up to time $t - 1$, and use this win rate as the prediction for round $t$. This algorithm has $N(N-1)/2$ parameters, and is the most expressive rating system. For detailed prediction rule and update rule of the aforementioned algorithms, see Appendix D. In this paper, we consider Elo, Glicko, Trueskill to be similar algorithms as they achieve qualitative similar results across almost all experiments we ran, despite the actual numbers being slightly different. We mainly compare Elo against more complex algorithms such as Elo2k and Pairwise. This is because the focus of this paper is on model misspecification. As we observe in a majority of our experiments, more complex algorithms have a clear advantage in reducing the model misspecification errors.

## 2.2 DATASETS

We utilize human gameplay data from online platforms for Chess, Scrabble, StarCraft, Hearthstone, and Go, as well as professional match records for ATP tennis (Sackmann, 2023) and Renju. Additionally, we incorporate human preference data from Chatbot Arena (Zheng et al., 2023), which can be alternatively viewed a game where LLM agents compete, with outcomes determined by human judgment.

## 3 EXPERIMENTS ON REAL-WORLD MATCHING DATA

In this section, we conduct experiments on real-world datasets. Surprisingly, we find that most games deviate significantly from the assumptions of the BT model and stationarity, raising questions on the reliability of Elo. Despite these deviations, Elo frequently outperforms more complex rating systems, such as mElo and pairwise models, which are designed to account for non-BT components in the data, particularly in terms of win rate prediction.

## 3.1 Real-world games are neither BT nor stationary

In the Elo rating update rule (2), $\sigma(\theta[i] - \theta[j])$ represents the predicted win probability of player i against player j . This prediction relies on the assumption that the underlying data follows the Bradley-Terry (BT) model. However, whether real-world data truly follows a stationary BT model remains unverified. In this section, we conduct a likelihood ratio test on real-world datasets to examine the hypothesis that real-world game outcomes are generated by the BT model. Our results indicate that, across all examined datasets, the hypothesis is rejected, suggesting that real-world data does not follow the BT model. Furthermore, we provide evidence that both matchmaking and player skill exhibit non-stationarity in real-world games. These findings suggest that model misspecification widely exists when applying Elo to real-world data.

**Rejecting BT on real-world dataset**  Note that the Bradley-Terry model can be equivalently written as a logistic regression model, where the parameter $\theta$ is $N$-dimensional, and every game has a feature vector $x_t := \mathbf{e}[i_t] - \mathbf{e}[j_t] \in \mathbb{R}^N$. We randomly split $[T]$ into $\mathcal{T}_{\text{train}}$ and $\mathcal{T}_{\text{test}} = [T] \setminus \mathcal{T}_{\text{train}}$. Then the logistic regression loss on the test set is defined as

$$\mathcal{L}_{\text{test}}(\theta) = - \sum_{t \in [T]} \left[ o_t \ln(\sigma(\theta^\top x_t)) + (1 - o_t) \ln(1 - \sigma(\theta^\top x_t)) \right].$$

Next, we *augment* the logistic model by adding two additional parameters $\alpha \in \mathbb{R}^2$, and a two dimensional feature $g_t \in \mathbb{R}^2$ for every game. In practice, $g_t$ is constructed using the training set $\mathcal{T}_{\text{train}}$ (see Appendix B for details, where two different augmentation strategies are provided). Define the negative log likelihood of the augmented model as

$$\widetilde{\mathcal{L}}_{\text{test}}([\theta; \alpha]) = - \sum_{t \in \mathcal{T}_{\text{test}}} \left[ o_t \ln(\sigma(\theta^\top x_t + \alpha^\top g_t)) + (1 - o_t) \ln(1 - \sigma(\theta^\top x_t + \alpha^\top g_t)) \right].$$

If dataset is indeed realizable by a BT model with true scores $\theta^\star$, the augmented model is also realizable with $[\theta^\star; \mathbf{0}]$ as long as $g_t$ and $o_t$ are independent, because $\mathbb{E}[o_t | i_t, j_t, g_t] = \sigma(\theta^\star[i_t] - \theta^\star[j_t])$. Therefore, we can test the BT model by testing the null hypothesis $H_0 : \alpha = 0$. We employ the standard likelihood ratio test, which uses the log-likelihood ratio statistic:

$$\Lambda := 2 \left[ \min_{\theta \in \mathbb{R}^N} \mathcal{L}_{\text{test}}(\theta) - \min_{\theta \in \mathbb{R}^N, \alpha \in \mathbb{R}^2} \widetilde{\mathcal{L}}_{\text{test}}([\theta; \alpha]) \right].$$

By Wilks' Theorem (Wilks, 1938; Sur et al., 2019), under the null hypothesis that the real-world dataset is generated by Bradley-Terry model, $\Lambda$ is asymptotically distributed as a chi-square distribution with two degrees of freedom. This allows us to compute the $p$-value, which is the probability that the test statistic occurs under the null hypothesis due to pure chance. We compute the log-likelihood ratio statistic $\Lambda$ for eight real-world datasets and report the corresponding $p$-values (see Table 1). It can be seen that we can reject the null hypothesis, namely realizability of the Bradley-Terry model, with extremely high confidence, for all eight datasets.

**Matchmaking and player skills are non-stationary**  Additional observations that we draw from real-world datasets are the existence of non-stationary matchmaking and player's skills. We postpone details to Appendix C. These phenomena suggest that the real world games are non-BT and non-stationary. Consequently, viewing Elo rating as fitting a underlying BT-model might not be appropriate.

## 3.2 Elo achieves good performance under model misspecification

Section 3.1 establishes that real-world games do not follow a stationary BT model, highlighting model misspecification in the applicaton of the Elo rating system. This raises important concerns regarding Elo's reliability in practical settings. In particular, it prompts the question of whether more sophisticated rating algorithms, such as Elo2k or Pairwise, which may better capture the underlying game distributions, could yield improved predictive performance. However, we examine the prediction accuracy for the next game outcome of various online algorithms in real-world datasets, and surprisingly find that despite the model misspecification, "Elo-like" algorithms still achieve strong predictive performance, outperforming complex algorithms even in some non-BT datasets. For each dataset, we compute the cumulative loss $\frac{1}{T} \mathcal{L}_T$ for Elo, Elo2k (with $k = 4$), Glicko, TrueSkill,

and Pairwise.[1] The results, summarized in Table 1, show that in several real-world datasets, including `Renju`, `Chess`, `Tennis`, `Scrabble`, `StarCraft` and `Go`, Elo and "Elo-like" rating outperform more complexity rating systems such as Elo2k and Pairwise.

# 4 UNDERSTAND ELO UNDER MISSPECIFICATION

The findings in Section 3.2 that the "Elo-like" algorithms outperform more complexity rating systems in some non-BT datasets, underscore the importance of adopting a new perspective on Elo (and other online algorithms), moving beyond the traditional view that Elo is merely a parameter estimation tool for the BT model.

In this section, we will explain this unexpected phenomenon through three key perspectives. First, we view game rating through the lens of regret minimization in online optimization. Specifically, Elo can be reinterpreted as an instance of online gradient descent under convex loss, which provides no-regret guarantees even in misspecified and non-stationary settings. Second, further synthetic experiments on non-BT and non-stationary datasets show that the "sparsity" of dataset is a critical factor in the performance of algorithms, driven by a trade-off between model misspecification error and regret. Finally regarding the ranking performance, we show that the pairwise ranking performance is strongly correlated with prediction performance, though Elo should not be blindly trusted since it can fail to produce consistent total orderings even in transitive datasets.

## 4.1 NEW LENS VIA REGRET MINIMIZATION

In this section, we will view game rating through the lens of regret minimization in online optimization. We will adapt the framework of Online Convex Optimization (OCO) to the online algorithms. To facilitate our presentation, we briefly introduce OCO, following Hazan et al. (2016)'s definition.

**Online Convex Optimization**   At iteration $t$, the online player chooses $x_t \in \mathcal{K}$ according to the information in steps $1, 2, \cdots, t-1$. After the player has committed to this choice, a cost function $f_t \in \mathcal{F} : \mathcal{K} \to \mathbb{R}$ is revealed. Here, $\mathcal{F}$ is the bounded family of cost functions available to the adversary. The cost incurred by the online player is $f_t(x_t)$, the value of the cost function for the choice $x_t$. Let $T$ denote the total number of game iterations. The regret is defined as

$$\text{Regret}_T := \sum_{t=1}^{T} f_t(x_t) - \min_{x \in \mathcal{K}} \sum_{t=1}^{T} f_t(x),$$

that is, the cumulative loss minus the optimal loss in hindsight.

It turns out that online rating algorithms can be evaluated under this framework. At each time $t$, let $f_t$ be the binary cross entropy loss function induced by the players $i_t$ and $j_t$ and the outcome $o_t$, and $x_t$ be the parameters updated by algorithms:

$$f_t(x_t) := -(o_t \ln p_t + (1 - o_t) \ln(1 - p_t)).$$

Here $p_t$ is actually related to the parameter $x_t$. Under this formulation, we have

$$\mathcal{L}_T = \text{Model misspecification error} + \text{Regret}_T. \tag{3}$$

From this equation, we can see that the cumulative loss consists of two components, the model misspecification error (optimal loss in hindsight) and the regret. The trade-off between these two terms will be illustrated in extensive experiments.

**Elo as online gradient descent**   For Elo update, $x_t := \theta_t \in \mathbb{R}^N$, is the parameter of the underlying BT model (the Elo score). $p_t := \sigma(\theta[i_t] - \theta[j_t])$ is the prediction. The gradient of $f_t$ is given by $\nabla_\theta f_t(\theta) = -(o_t - p_t)(e_{i_t} - e_{j_t})$. We can see that the Elo score update is actually online gradient descent with learning rate $\eta_t$ at each step $t$. Notice that $f_t$ is a convex function (one can refer to Appendix D for detail). Therefore we can apply the regret bound for online gradient descent under convex loss (Hazan et al., 2016, Theorem 3.1):

---
[1]The experimental details can be found in Appendix E.

**Theorem 1.** *For convex cost functions $\{f_t\}_{t=1}^{T}$ and convex set $\mathcal{K}$, online gradient descent with step sizes $\{\eta_t = \frac{D}{G\sqrt{t}}\}$ guarantee the following for all $T > 1$:*

$$Regret_T = \sum_{t=1}^{T} f_t(x_t) - \min_{x \in \mathcal{K}} \sum_{t=1}^{T} f_t(x) \leq \frac{3}{2} GD\sqrt{T},$$

*where $D$ is the upper bound on the diameter of $\mathcal{K}$, and $G$ is an upper bound on the norm of the subgradients of $f_t$ over $\mathcal{K}$.*

In the context of Elo update, since $\theta \in \mathbb{R}^N$, and in experiments we observe that $\|\theta\|_\infty \leq 5$, which means we can choose $D = 10\sqrt{N}$. For $G$, recall $\nabla_\theta f_t(\theta) = -(o_t - p_t)(e_{i_t} - e_{i_t})$, we have $G \leq \sqrt{2}$. Therefore we conclude that online Elo score update will have the following regret bound: $\frac{1}{T} Regret_T \leq C\sqrt{\frac{N}{T}}$ for some absolute constant $C$, with learning rate $\eta_t = \sqrt{\frac{N}{t}}$. Notice that this regret bound even holds under misspecified and non-stationary settings, which explains Elo's good performance in non-BT datasets, as long as the best BT model in hindsight provides a reasonable fit to the data.

We can also formulate the Elo2k update under online optimization framework as the following:

$$f_t(\theta) := -(o_t \ln p_t + (1 - o_t) \ln(1 - p_t)),$$

where $\theta = (U, V)$, where $U = (u_1, \cdots, u_N), V = (v_1, \cdots, v_N)$, $u_i, v_i \in \mathbb{R}^k$. The prediction $p_t = \sigma(u_{i_t}^T v_{j_t} - u_{j_t}^T v_{i_t})$. Then the Elo2k online update will be online gradient descent. However, the loss function is non-convex, therefore a general guarantee of OGD under Elo2k model is lacking.

### 4.2 Synthetic experiments: sparsity is critical

To further justify our interpretation of why Elo performs well even in non-BT datasets, in this section, we will conduct extensive synthetic experiments, as well as experiments on augmented real-world data. These experiments further show that the "sparsity" of the dataset plays a crucial role in the performance of algorithms.

**Synthetic experiments on non-BT datasets**   We begin with the scenario where the players' skills are stationary in the sense that $\mathbb{E}[o_t | i_t = i, j_t = j] = P_{ij}$ for some matrix $P \in \mathbb{R}^{N \times N}$. We consider the following two notions of the transitivity:

**Definition 1** (SST). *$P$ is strongly stochastic transitive (SST) with respect to ordering $\pi$ if $\pi(i) > \pi(j)$ implies $P_{ik} \geq P_{jk}$ for all $k \in [N]$.*

**Definition 2** (WST). *$P$ is weakly stochastic transitive (WST) with respect to ordering $\pi$ if $\pi(i) > \pi(j)$ implies $P_{ij} \geq \frac{1}{2}$.*

It is well-known that BT implies the SST condition, and SST further implies WST (see Appendix F.1 for details of the constructed $P$). For each of these types of $P$, we generate $P$ for $N = 1000$ and $N = 100$. For each instance of $P$, we generate $T = 10^5$ games following uniform matchmaking distribution, that is, for every $t \in [T]$, sample $i_t \sim \text{Uni}([N])$, then independently sample $j_t \sim \text{Uni}([N])$. For each algorithm, we choose the best hyperparameter (for details of choosing the best hyperparameter, see Appendix F.2), we plot the corresponding $\frac{1}{t}\mathcal{L}_t$ with respect to time step $t/N$ (Figure 1 for $N = 1000$ and Figure 3 for $N = 100$). The model misspecification error (optimal loss in hindsight) at time $T$ is also plotted for $N = 100$. We can see from Figure 1 and 3 that the effectiveness of rating algorithms is shaped by the interaction between data sparsity and model complexity. There is a trade-off between the regret and the model misspecification error: when the samples are sparse, i.e., $t$ is small, the dominating term in the cumulative loss will be the regret, Elo2k or Pairwise suffers from a huge regret. Under this scenario, Elo and its variants performs well due to its low regret, even though BT model is non-realizable. For dense regime, i.e., $t$ is large, the regret for both Elo and Elo2k will be closer to zero. Under this scenario, Elo2k or Pairwise may achieve superior performance when they achieve a lower misspecification error due to their greater model capacity.

**Non-trivial matchmaking and varying player strengths**   We further justify our regret-minimization framework through synthetic experiments under the scenario where the player strengths

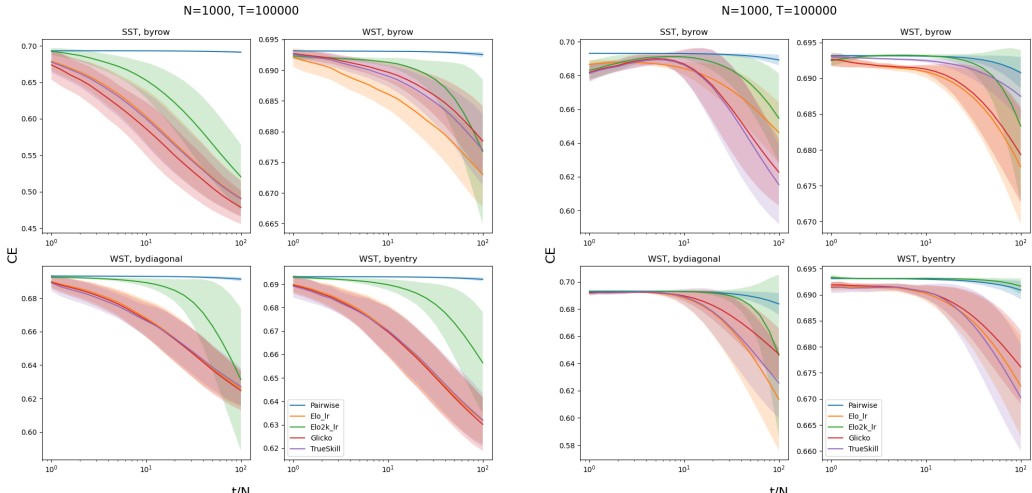

Figure 1: **Elo outperforms Elo2k in sparse datasets.** We can see that Elo consistently outperforms Elo2k for $0 \le t \le T$. One can also observe that when $t$ gets larger, Elo2k's performance is rapidly improving.

Figure 2: **Elo is more robust to non-stationary matchmaking and player strengths.** Compared with Figure 1, we can see that the performance of Elo2k deteriorates more than Elo.

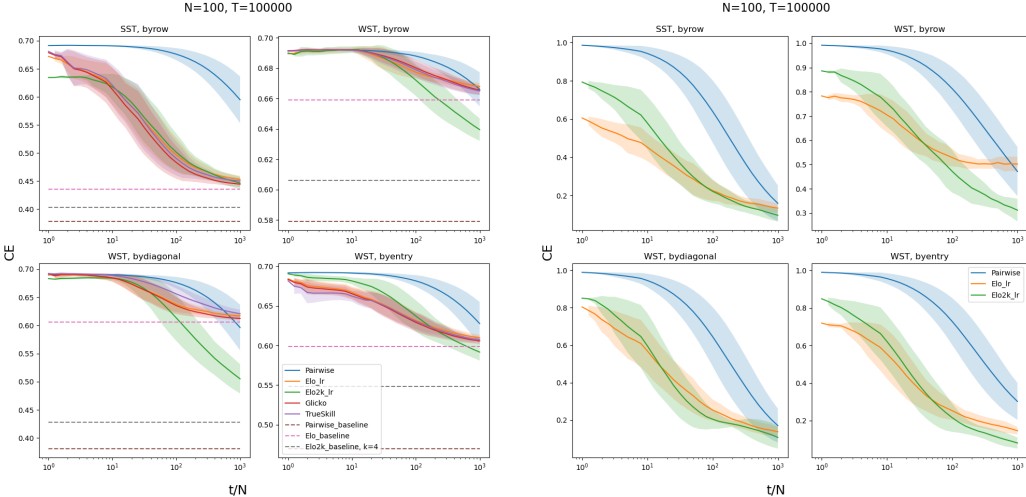

Figure 3: **Elo2k has the potential to outperform Elo in dense datasets.** When $t$ is large, the regret are close to zero for all algorithms, and Elo2k outperforms Elo when the gap between Elo and Elo2k in hindsight baselines are large.

Figure 4: **Pairwise ranking is consistent with prediction** Elo performs the best for small $t$, Elo2k performs well for moderate $t$, Pairwise performs the best for large $t$.

can vary and a non-trivial matchmaking exists. We plot the performance of each algorithm in non-stationary datasets ($N = 1000$) in Figure 2 (see Appendix F.3 for experiment details). Comparing Figure 1 with Figure 2, we can see that when non-trivial matchmaking exists and the player strength are varying, Elo still performs reasonably well, while Elo2k exhibits a significant deterioration in performance. This also justifies our finding: Elo as online gradient descent, is guaranteed to achieve a low regret, even under non-trivial matchmaking and non-stationary player strengths.

**Experiments on real-world data** Similar behaviors also appears in real-world datasets. Other than the real-world datasets examined in Section B, we also use `Blotto` and `AlphaStar` data

from Czarnecki et al. (2020), where we generate game data from the original payoff matrix. To create denser datasets, we augment datasets from Czarnecki et al. (2020) by simply creating identical copys. For each real-world dataset, we plot the corresponding $\frac{1}{t}\mathcal{L}_t$ for each algorithm with respect to time step $t/N$ (Figure 7). We can also see that sparsity plays a crucial role in those real-world (or augmented) datasets, as in the previous synthetic experiments. See Appendix E for details.

## 4.3 RANKING PERFORMANCE OF ELO

Besides prediction, ranking is another important aspect that users consider when utilizing rating algorithms. There are two types of ranking: (1) for general $P$, we can consider the pairwise ranking, i.e., for each pair $(i,j) \in [N] \times [N]$, there is a ranking between $i, j$ that is induced by $P_{ij}$. (2) for transitive $P$, there exists a ground truth ranking $\pi$ induced by the transitivity. In this subsection, we will show that for pairwise ranking, the ranking performance is strongly correlated to prediction performance. Elo rating, achieves good performance of pairwise ranking in sparse regimes. However Elo should not be blindly trusted, since for the total ordering, Elo may not always give a consistent ranking, even in transitive datasets.

**Good prediction gives good pairwise ranking** Regarding the pairwise ranking, it is natural to conjecture that pairwise ranking performance is correlated with the prediction performance, and our synthetic experiments justify this claim. We consider the same setup as the previous synthetic experiments for prediction (Section 4.2), and we calculated the pairwise ranking consistency for each algorithm at each time step: at time $t$, an algorithm can actually give a prediction $\hat{P}_{ij}$ for every pair $(i,j) \in [N] \times [N]$. We calculate the following quantity: $\tau := \frac{1}{N(N-1)} \sum_{i \neq j} (\mathbb{1}[P_{ij} > 0.5]\mathbb{1}[\hat{P}_{ij} < 0.5] + \mathbb{1}[P_{ij} < 0.5]\mathbb{1}[\hat{P}_{ij} > 0.5])$. Lower the value, more consistent the pairwise ranking. We plot $\tau$ against $t/N$ for Elo, Elo2k and Pairwise in Figure 4. e can see that the ranking performance is strongly correlated with the prediction performance. To be specific, similar to the prediction accuracy, in most sparse regimes, Elo performs well in pairwise ranking. However in denser regimes, algorithms based on more complex models, such as Elo2k, may show advantage on pairwise ranking.

**Elo might not give consistent total ordering even for transitive models** For transitive models, we can consider the total ordering induced by the transitivity. Elo rating, is still able to give a total ordering based on the score of each player. We will show that even though Elo can give a consistent total ordering under uniform matchmaking, it can not be blindly trusted as it may fail under arbitrary matchmaking.

From theoretical perspective, we consider the regime where $T$ goes to infinity. In this regime, by the no regret nature of OGD, one can see that the online Elo update will finally converge to the offline Elo solution $\arg\min_{\theta \in \mathcal{K}} \frac{1}{T} \sum_{t=1}^{T} f_t(\theta)$. Also we can see that when $T \to \infty$, $\frac{1}{T} \sum_{t=1}^{T} f_t(\theta)$ converge to its population counterpart. Therefore we will consider $\theta^{\mathrm{mle}}$, the population MLE for BT model. We have the following result:

**Theorem 2.** *Under uniform matchmaking, $\theta^{\mathrm{mle}}$ gives identical ranking as $\overline{P}$, where*

$$\overline{P}[i] := \frac{\sum_{t=1}^{T}(\mathbb{1}[i_t=i]o_t + \mathbb{1}[j_t=i](1-o_t))}{\sum_{t=1}^{T}(\mathbb{1}[i_t=i] + \mathbb{1}[j_t=i])}$$

*is the average win rate for player $i$.*

the formal statement and proof is deferred to Appendix G.1. Notice that under SST models, the ground truth ranking is identical to the ranking given by average win rate. Therefore this theorem shows that under SST model, Elo recovers the true ranking when $T$ goes to infinity, under uniform matchmaking.

However, when the underlying model is WST, the ranking induced by average win rate may not be correct, therefore Elo is not guaranteed to be consistent. Moreover, when the matchmaking is *arbitrary*, Elo score can produce inconsistent rankings for SST instances even when $\eta \to 0$ and $T \to \infty$. We also show through a synthetic experiment that even in the case where only ranking among players that have confidently separated Elo scores are considered, Elo still may not give consistent ranking. For these results, see Example 1 in Appendix G for detail. This suggest that although Elo can give good ranking results in many regimes, it can not be blindly trusted.

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

## A  DATASET DESCRIPTION

**Renju**   We use the RenjuNet dataset [2], where $N = 5013$ and $T = 124948$.

**LLM Arena**   We use the Chatbot Arena [3] dataset, where $N = 129$ LLMs are evaluated through $T = 1493621$ battles.

**Chess**   We use online standard chess matches on the Lichess open database [4] in the year of 2014, which covers $N = 184920$ players and $T = 11595431$ games.

**Mixedchess**   We use online matches on the Lichess database [5], for the following variants of chess: Antichess, Atomic, Chess960, Crazyhouse, Horde, King of the Hill, Racing Kings and Three-check. We collect all the matches from 2014 to 2024, and combine them together to form a large dataset with $N = 2467134$ and $T = 115525041$. We apply filtering [6] with threshold = 10000, and create `mixedchess-dense`, where $N = 2862$ ,$T = 11791126$.

**Tennis**   We make use of an online repository created by Jack Sackmann [7]. We used all $T = 190230$ games between 7245 players.

**StarCraft**   We downloaded match records from human players from Aligulac [8]. $N = 22056$ and $T = 427042$.

**Scrabble**   We used the raw data provided in Roeder (2017), which are scraped from http://cross-s$^{(2)}$tables.com. $N = 15374$ and $T = 1542642$.

**Go**   We use the Online Go Server (OGS) database [9], which contains $T = 12876823$ games between 426105 players. We also filter the dataset with threshold 5000, to get `go-dense` with $N = 480$ and $T = 516343$.

**Hearthstone**   We use the Deck Archetype Matchup data scraped from https://metastats. net/hearthstone/archetype/matchup/. $N = 27$ and $T = 62453$.

**Blotto**   We use the 5,4-Blotto and 10,5-Blotto data from Czarnecki et al. (2020). The original data is a payoff matrix between different strategies, whose entries are between $[-1, 1]$. For strategies $i$ and $j$, let the payoff for $i$ against $j$ be $r_{ij}$, we let $p_{ij} := 0.5(r_{ij} + 1)$. For `5,4-Blotto` and `10,5-Blotto`, we create games $(i_t, j_t, o_t) := (i, j, p_{ij})$ for each $(i, j)$ where $i \neq j$. For `5,4-Blotto-sparse`, we create $0 - 1$ game results by drawing $o_i j \sim \text{Ber}(p_i j)$ and create games $(i_t, j_t, o_t) := (i, j, o_i j)$. For `5,4-Blotto-dense`, we create $0 - 1$ game results by drawing 10 independent $o_i j \sim \text{Ber}(p_i j)$ and create 10 games $(i_t, j_t, o_t) := (i, j, o_i j)$ for each pair of $(i, j)$. `10,5-Blotto-sparse` and `10,5-Blotto-dense` are similarly created. $N = 56, T = 3080$ for `5,4-Blotto` and `5,4-Blotto-sparse`. $N = 56, T = 30800$ for `5,4-Blotto-dense`. $N = 1001, T = 1001000$ for `10,5-Blotto` and `10,5-Blotto-sparse`. $N = 1001, T = 10010000$ for `10,5-Blotto-dense`.

**AlphaStar**   We use the AlphaStar data from Czarnecki et al. (2020). The creation procedure for `AlphaStar`, `AlphaStar-sparse`, `AlphaStar-dense` are the same as in Blotto. For all the three dataset, $N = 888$. $T = 787656$ for the first two datasets, and $T = 7876560$ for the last one.

---

[2]https://www.renju.net/game/
[3]https://chat.lmsys.org/
[4]https://database.lichess.org/
[5]https://database.lichess.org/
[6]To create denser dataset, we conduct filtering on mixedchess and go. Our filtering method is: for a given threshold, we delete all the players that plays less than this threshold. Then we only consider the remaining players and the games played between them.
[7]https://github.com/JeffSackmann/tennis_atp
[8]http://aligulac.com/about/db/
[9]https://github.com/online-s$^{(2)}$go/goratings

## B  DETAILS OF THE LIKELIHOOD RATIO TEST

We explain in this section the details of our likelihood ratio tests.

**High-dimensional scenario**  For high-dimensional logistic regression, Sur et al. (2019) showed that $\Lambda$ is asymptotically distributed as a scaled chi-square distribution if $T/N = O(1)$. We applied the correction suggested by Sur et al. (2019) by computing the $p$-value conservatively with $1.25\chi_2^2$. This factor is computed when the number of samples is 5 times the model dimension, although the number of samples is at least 19 times the model dimension in our datasets.

**Symmetrization**  Before performing the tests, we reversed the order of the two players (and flipped the game outcome) for every game with probability 0.5 to eliminate first-move advantage (or disadvantage), which is well-documented (Elo, 1978) and not the focus of this work. In other words, we actually test the following weaker version of Bradley-Terry model:

$$\frac{1}{2} \left( \mathbb{E}\left[o_t | i_t, j_t\right] + \mathbb{E}\left[1 - o_t | j_t, i_t\right] \right) = \sigma \left( \theta^\star[i_t] - \theta^\star[j_t] \right).$$

### B.1  FEATURE AUGMENTATION

Here we use two feature augmentation strategies. For both strategies, we split every dataset (indexed by $[T]$) randomly into equally sized $\mathcal{T}_{\text{train}}$ and $\mathcal{T}_{\text{test}}$. The augmented feature is constructed using $\mathcal{T}_{\text{train}}$, and the test is performed on $\mathcal{T}_{\text{test}}$. The test results are presented in Table 2.

**Feature based on Elo score**  The first strategy is based on Elo score. To be specific, we fit a regularized logistic regression model via

$$\theta_{\text{train}} \leftarrow \arg\min_\theta \sum_{t \in \mathcal{T}_{train}} \ell_t(\theta) + \frac{\lambda}{2} \|\theta\|^2,$$

where we chose $\lambda = 10.0$. Then the augmented features for match $t$ is given by

$$g_t := [\theta_{\text{train}}[i_t], \theta_{\text{train}}[j_t]],$$

for every $t$ in the test set. Under the Bradley-Terry model, the original logistic regression $\mathcal{L}_{\text{test}}(\theta)$ already has sufficient information to predict $o_t$, so adding the score computed on an independent training set cannot help prediction (up to random noise).

**Feature based on Elo2k**  The second strategy is based on fitting Elo2k model (Bertrand et al., 2023) on dense datasets. We considered the loss

$$\hat{\mathcal{L}}(u, v) := \sum_{t \in \mathcal{T}_{\text{train}}} \left[ -o_t \ln(\sigma(\hat{p})) - (1 - o_t) \ln(1 - \sigma(\hat{p})) \right],$$

where $\hat{p} := u[i_t]v_[j_t] - u[j_t]v[i_t]$. The loss is optimized with gradient descent with early stopping. We then define

$$g_t := [u[i_t]v[j_t], u[j_t]v[i_t]]$$

as the augmented feature for game $t$. This method does not apply to sparse datasets as it requires a dense dataset for the learning of $u$ and $v$.

### B.2  IMPLEMENTATION

All logistic regressions are implemented with JAX and optimized via L-BFGS.

### B.3  AN ADDITIONAL MARTINGALE TEST

Although the previous test used randomly sampled $\mathcal{T}_{\text{train}}$, it still needs to assume that the features $\{x_t\}_{t \in \mathcal{T}_{\text{train}}}$ are independent with $\{y_t\}_{t \in \mathcal{T}_{\text{test}}}$. However, there is a concern that this may not be true

| Dataset | Test Statistic (Feature based on Elo) | $p$-value | Test Statistic (Feature based on Elo2k) | $p$-value |
|---|---|---|---|---|
| Renju | 150.0 | $< 10^{-10}$ | - | - |
| Chess | 2020.1 | $< 10^{-10}$ | - | - |
| Tennis | 37.3 | $< 10^{-4}$ | - | - |
| Scrabble | 142.2 | $< 10^{-10}$ | - | - |
| StarCraft | 775.8 | $< 10^{-10}$ | - | - |
| Go | 193411.2 | $< 10^{-10}$ | - | - |
| LLM Arena | 81.627 | 0.443 | 73.1 | $1 \times 10^{-3}$ |
| Hearthstone | 1.447 | 0.485 | 49.0 | $< 10^{-4}$ |

Table 2: Summary of two Likelihood ratio tests based on different features

| Dataset | LR Test Statistic ($\eta = 0.01$) | $p$-value | LR Test Statistic ($\eta = 0.08$) | $p$-value |
|---|---|---|---|---|
| Renju | 3.66 | 0.23 | 226.92 | $< 10^{-10}$ |
| Chess | 6622.52 | $< 10^{-10}$ | 27908.30 | $< 10^{-10}$ |
| Tennis | 524.77 | $< 10^{-10}$ | 3571.70 | $< 10^{-10}$ |
| Scrabble | 174.52 | $< 10^{-10}$ | 3058.76 | $< 10^{-10}$ |
| StarCraft | 5.19 | 0.12 | 261.08 | $< 10^{-10}$ |
| Go | 52931.15 | $< 10^{-10}$ | 117318.3 | $< 10^{-10}$ |
| LLM Arena | 872.28 | $< 10^{-10}$ | 819.05 | $< 10^{-10}$ |
| Hearthstone | 69433.52 | $< 10^{-10}$ | 82005.34 | $< 10^{-10}$ |

Table 3: Summary of martingale-based Likelihood ratio test

if adaptive matchmaking is used – in that case, information about the test set labels $\{y_t\}_{t \in \mathcal{T}_{\text{test}}}$ may be leaked through features of future games. [10]

To address this concern, we consider yet another method to construct $g_t$: by using the online Elo rating up until this point. This enables us to relax the assumption of independence to the assumption that the noise sequence

$$\mathbb{E}[o_t | i_t, j_t] - \sigma(\theta^\star[i_t] - \theta^\star[j_t])$$

is a martingale. This would enable us to model adaptive matchmaking.

Specifically, define

$$g_t = [\theta_t[i_t], \theta_t[j_t]],$$

where $\theta$ is computed using the past $t - 1$ matches with learning rate $\eta$. We can then proceed to compute the likelihood ratio statistic $\Lambda$ as in the previous tests. The distribution of $\Lambda$ would still be asymptotically $\chi_2^2$ for martingale noise (see e.g. Kedem & Fokianos (2005, Theorem 1.5.1)) The test results are presented in Table 3.

### B.4 ANALYSIS OF ALL THE THREE TEST RESULTS

**Result for different augmented features** The test results of two different strategies of augmenting features are presented in Table 2. We can see that in sparse datasets (i.e., datasets excepts LLM Arena and Hearthstone), the test that uses features based on Elo, rejects the null hypothesis with high confidence. While for dense datasets such as LLM Arena and Hearthstone, test that uses features based on Elo2k (which is more suitable for denser datasets), reject the hypothesis with high confidence.

**Martingale based test reject the null with high confidence** We report the test results of martingale based test in Table 3. We find that by using two learning rates ($\eta = 0.01$ and $0.08$), we can reject the null hypothesis that BT is realizable with extremely high confidence without assuming independence.

In summarize, we designed three different testing strategies to reflect different aspects of real world datasets violating the BT model assumption. The results indicate that in real world, the data

---

[10]Regarding AlphaStar, 5, 4-Blotto and 10, 5-Blotto, recall that we construct the dataset according to a payoff matrix, therefore no adaptive matchmaking is used. We do not need to further test these datasets.

distribution is likely non-BT. In the main text, we report the testing results of non-martingale based test in Table 1, where for `LLM Arena` and `Hearthstone` we report the results with Elo2k based features, for other six datasets we report the results with Elo based features

## C    NON-STATIONARY MATCHMAKING AND PLAYER SKILLS IN REAL DATASETS

Another observation that we draw from real-world datasets is the existence of non-trivial matchmaking. We computed the correlation coefficient between $\{\theta_{\mathrm{train}}[i_t]\}_{t \in \mathcal{T}_{\mathrm{test}}}$ and $\{\theta_{\mathrm{train}}[j_t]\}_{t \in \mathcal{T}_{\mathrm{test}}}$, and found significant positive correlation for most datasets (see Table 4). In other words, in many real datasets, stronger (higher-rated) players are matched with stronger opponents. We visualize the matchmaking in `chess` in Fig. 5. Indeed, the Elo score of the two players are highly correlated, and most games are played between two players within $20\%$ in terms of the percentile difference based on their Elo scores. Since the Elo score may vary from time to time, the matchmaking distribution should not be considered as stationary.

| Dataset | Matchmaking Test | $p$-value |
|---|---|---|
| Renju | 0.36 | $< 10^{-10}$ |
| Chess | 0.40 | $< 10^{-10}$ |
| Tennis | 0.19 | $< 10^{-10}$ |
| Scrabble | 0.57 | $< 10^{-10}$ |
| StarCraft | 0.46 | $< 10^{-10}$ |
| Go | 0.29 | $< 10^{-10}$ |
| LLM Arena | 0.37 | $< 10^{-10}$ |
| Hearthstone | -0.07 | $< 10^{-10}$ |

Table 4: Summary of real world datasets matchmaking hypothesis testing results.

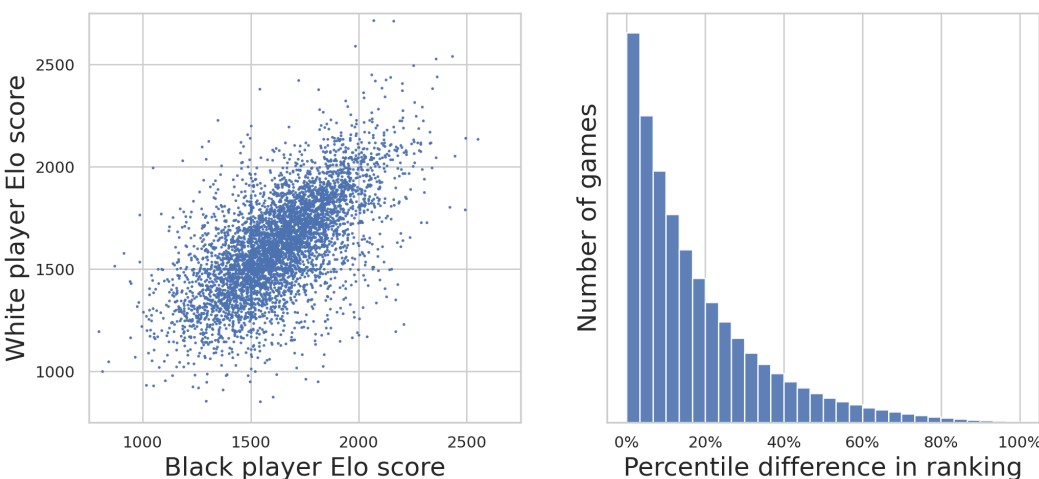

Figure 5: **Matchmaking in `chess` dataset. L:** scatter plot of Elo score of the two players for each game, down-sampled for clarity; **R:** histogram for the percentile ranking difference of two players.

**Bootstrap Experiments**    Another evidence of matchmaking comes from the nonstationarity of gradients. If the distribution of $\{(i_t, j_t, o_t)\}$ is exchangeable, we can permute the order of the games randomly and the resulting Elo score $\theta^{\mathrm{bootstrap}}$ should be identically distributed. We can therefore detect nonstationarity by comparing $\theta^{\mathrm{elo}}$ with the distribution of $\theta^{\mathrm{bootstrap}}$.

We compute the Elo score on 100 independent permutations in the each dataset. The average of these samples is called the bootstrap average, and denoted by $\bar{\theta}^{\mathrm{bootstrap}}$.

The results for `chess` is presented in Fig. 6. It can be seen that $\theta^{\mathrm{elo}}$, the Elo score computed with the original order of gradients, is a significant outlier and is not identically distributed with $\theta^{\mathrm{bootstrap}}$ with high probability ($p = 0.01$ via the permutation test).

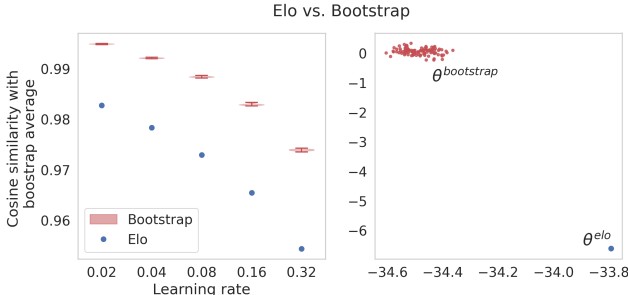

Figure 6: Elo score vs. bootstrap Elo scores in `chess`. **Left:** cosine similarity to the mean of $\theta^{\mathrm{bootstrap}}$; **Right:** visualization of $\theta^{\mathrm{elo}}$ vs. $\theta^{\mathrm{bootstrap}}$ via SVD for $\eta = 0.02$.

**Varying player strengths**   Other than matchmaking, we also want to point out that the player's strength may not be stationary. It is common that for a pair of players, for example, in tennis, their head-to-head game results can change dramatically over time.

These phenomenons suggest that, in real world games, both matchmaking and players' behaviour are not stationary and non-BT. Therefore, viewing Elo rating as fitting a underlying BT-model might not be appropriate.

# D   ONLINE RATING ALGORITHMS

In this paper, we will investigate the performance of the following algorithms: Elo, Glicko, Elo2k and Pairwise.

**Elo**   Elo rating gies the prediction $p_t := \sigma\left(\theta_t[i_t] - \theta_t[j_t]\right)$. Initially $\theta_0[i] = 0$ for every $i \in [N]$. The update rule is:

$$\begin{cases} \theta_{t+1}[i_t] & \leftarrow \theta_t[i_t] + \eta_t\left(o_t - p_t\right), \\ \theta_{t+1}[j_t] & \leftarrow \theta_t[j_t] - \eta_t\left(o_t - p_t\right). \end{cases} \tag{4}$$

Here $\sigma = 1/(1 + e^{-x})$ is the logistic function. $\theta_t \in \mathbb{R}^N$ is the *rating*, or *score*, for the $N$ players at time $t$. Customarily, the reported rating is multiplied by a constant $C = \frac{400}{\ln 10}$. The learning rate $\eta_t$ is often chosen to be a fixed value $\eta$ between $10/C \approx 0.06$ and $40/C \approx 0.23$. In our experiments, we choose $\eta_t$ according to the following decaying learning rate scheme: $\eta_t = \sqrt{\frac{aN}{t+b}}$, where $a, b$ are chosen to ensure the learning rate will not be too large at the beginning, and still large enough for achieving a good prediction accuracy when $t$ is large. For details see F.2.

Elo update can be understand as online gradient descent, as we described in Section 4.1. Also, we can show that $f_t$ is convex:

$$\begin{aligned} \nabla_\theta f_t(\theta) &= -\nabla_\theta(o_t \ln p_t + (1 - o_t)\ln(1 - p_t)) \\ &= -(o_t\frac{1}{p_t}\nabla_\theta p_t + (1 - o_t)\frac{1}{1 - p_t}(-\nabla_\theta p_t)) \\ &= -\frac{o_t - p_t}{p_t(1 - p_t)}\nabla_\theta p_t \\ &= -(o_t - p_t)(\boldsymbol{e}_{i_t} - \boldsymbol{e}_{j_t}). \end{aligned}$$

$$
\begin{aligned}
\nabla_\theta^2 f_t(\theta) &= \nabla_\theta(\nabla_\theta f_t(\theta)) \\
&= -\nabla_\theta((o_t - p_t))(\boldsymbol{e}_{i_t} - \boldsymbol{e}_{j_t}) \\
&= -(\boldsymbol{e}_{i_t} - \boldsymbol{e}_{j_t})(\nabla_\theta(o_t - p_t))^T \\
&= p_t(1 - p_t)(\boldsymbol{e}_{i_t} - \boldsymbol{e}_{j_t})(\boldsymbol{e}_{i_t} - \boldsymbol{e}_{j_t})^T \\
&\succeq 0.
\end{aligned}
$$

Details can be seen in Section 4.1.

**Glicko**  Glicko (Glickman, 1995) assumes each player has a rating $\theta$ and a "ratings deviation" $v$. The initial $\theta$ of each player is set to be 1500, and we set initial $v$ to be $35, 100$ or $350$. The prediction $p_t := \sigma(\frac{\ln 10}{400} g(\sqrt{v_t[i_t]^2 + v_t[j_t]^2})(\theta_t[i_t] - \theta_t[j_t]))$, where $\sigma$ is the logistic function. The update rule of the parameters is

$$
\begin{cases}
\theta_{t+1}[i_t] & \leftarrow \theta_t[i_t] + \frac{\ln 10}{400}(\frac{1}{v_t[i_t]^2} + \frac{1}{d(i_t,j_t)^2})^{-1} g(v_t[j_t])(o_t - \widetilde{p}(i_t, j_t)), \\
\theta_{t+1}[j_t] & \leftarrow \theta_t[j_t] + \frac{\ln 10}{400}(\frac{1}{v_t[j_t]^2} + \frac{1}{d(j_t,i_t)^2})^{-1} g(v_t[i_t])(1 - o_t - \widetilde{p}(j_t, i_t)), \\
v_{t+1}[i_t] & \leftarrow \sqrt{(\frac{1}{v_t[i_t]^2} + \frac{1}{d(i_t,j_t)^2})^{-1}}, \\
v_{t+1}[j_t] & \leftarrow \sqrt{(\frac{1}{v_t[j_t]^2} + \frac{1}{d(j_t,i_t)^2})^{-1}},
\end{cases}
\tag{5}
$$

where $g(x) := (1 + \frac{3(\frac{\ln 10}{400})^2 x^2}{\pi^2})^{-\frac{1}{2}}$, $\widetilde{p}(i_t, j_t) := v(\frac{\ln 10}{400} g(v_t[i_t])(\theta_t[i_t] - \theta_t[j_t]))$, and $d(i_t, j_t)^2 := ((\frac{\ln 10}{400})^2 g(v_t[j_t]^2)\widetilde{p}(i_t, j_t)(1 - \widetilde{p}(i_t, j_t)))^{-1}$.

**TrueSkill**  TrueSkill (Dangauthier et al., 2007) assumes each player has an average skill $\theta$ and a degree of uncertainty $v$, similar to Glicko. The difference is that TrueSkill use a Gaussian function for prediction, rather than logistic: the prediction $p_t := \frac{1}{c\sqrt{2}}\Phi(\sqrt{2}(\theta_t[i_t] - \theta_t[j_t]))$, where $\Phi$ is the CDF for standard normal distribution, and $c_t = \sqrt{2\beta^2 + v_t[i_t]^2 + v_t[j_t]^2}$ is the overall variance. In our experiments, we set $\beta$ to be $0.2, 0.8$ or $1.0$, and the initial $v^2$ for every player is set to be $4\beta^2$ by default.

The update rule for the parameters is

$$
\begin{cases}
\theta_{t+1}[i_t] & \leftarrow \theta_t[i_t] + (2o_t - 1)\frac{v_t[i_t]^2}{c_t^2} v(\frac{(\theta_t[i_t] - \theta_t[j_t])(2o_t - 1)}{c_t}), \\
\theta_{t+1}[j_t] & \leftarrow \theta_t[j_t] - (2o_t - 1)\frac{v_t[j_t]^2}{c_t^2} v(\frac{(\theta_t[i_t] - \theta_t[j_t])(2o_t - 1)}{c_t}), \\
v_{t+1}[i_t] & \leftarrow v_t[i_t]\sqrt{1 - \frac{v_t[i_t]^2}{c_t^2} w(\frac{(\theta_t[i_t] - \theta_t[j_t])(2o_t - 1)}{c_t})}, \\
v_{t+1}[j_t] & \leftarrow v_t[j_t]\sqrt{1 - \frac{v_t[j_t]^2}{c_t^2} w(\frac{(\theta_t[i_t] - \theta_t[j_t])(2o_t - 1)}{c_t})}
\end{cases}
\tag{6}
$$

where $v(x) := \frac{\phi(x)}{\Phi(x)}$ ($\phi$ is the pdf of standard Gaussian), $w(x) := v(x)(v(x) + x)$.

**Elo2k**  If we generalize Elo score by rating every player with a vector instead of scalar (see Balduzzi et al. (2018) and Bertrand et al. (2023)), we get Elo2k. The parameter for the algorithm is $\theta = (U, V)$, where $U = (U[1], \cdots, U[N]), V = (V[1], \cdots, V[N])$, $U[i], V[i] \in \mathbb{R}^k$. The prediction $p_t := \sigma(U[i_t]^T V[j_t] - U[j_t]^T V[i_t])$. In this paper, we initially choose each element of $U$ (or $V$) from Uniform$([0, 0.1])$. The update rule is given by taking the gradient of $U, V$, i.e.,

$$
\begin{cases}
U_{t+1}[i_t] & \leftarrow U_t[i_t] + \eta_t\,(o_t - p_t)\,V_t[j_t], \\
U_{t+1}[j_t] & \leftarrow U_t[j_t] - \eta_t\,(o_t - p_t)\,V_t[i_t], \\
V_{t+1}[i_t] & \leftarrow V_t[i_t] - \eta_t\,(o_t - p_t)\,U_t[j_t], \\
V_{t+1}[j_t] & \leftarrow V_t[j_t] + \eta_t\,(o_t - p_t)\,U_t[i_t]
\end{cases}
\tag{7}
$$

In our experiments, we choose $\eta_t$ according to the following decaying learning rate scheme: $\eta_t = \sqrt{\frac{aN}{t+b}}$.

**Pairwise** A very natural algorithm is that we compute the pairwise win rate $P_t[i, j]$ for each pair of players $(i, j)$, and the prediction $p_t = P_t[i_t, j_t]$. This algorithm has $N(N-1)/2$ parameters. To ensure the prediction will not be affected dramatically by a single game result, we will regularize it as the following. The update rule is given by

$$\begin{cases} P_t[i_t, j_t] & \leftarrow \frac{5 + \#\{\text{games that } i_t \text{ wins } j_t \text{ until time } t\}}{10 + \#\{\text{games that } i_t \text{ plays with } j_t \text{ until time } t\}}, \\ P_t[j_t, i_t] & \leftarrow \frac{5 + \#\{\text{games that } j_t \text{ wins } i_t \text{ until time } t\}}{10 + \#\{\text{games that } j_t \text{ plays with } i_t \text{ until time } t\}} \end{cases} \tag{8}$$

# E DETAILS OF REAL-WORLD DATA EXPERIMENTS

For each dataset, we evaluate the performance of Elo, Elo2k (with $k = 4$), Glicko, TrueSkill, and Pairwise, plotting the cumulative loss $\frac{1}{t}\mathcal{L}_t$ over "normalized" time $t/N$. For each algorithm, we choose the best hyperparameter (see Appendix F.2). We also plot the in hindsight baselines at time $T$ ($\min_{x \in \mathcal{K}} \frac{1}{T} \sum_{t=1}^{T} f_t(x)$) of BT model and Elo2k model. The results are presented in Figure 7. We can observe that for several real-world datasets, including chess, go, renju, tennis, scrabble and StarCraft, Elo and its variants (TrueSkill and Glicko) outperform algorithms based on more complex models such as Elo2k and Pairwise. Namely, Elo consistently exhibits a lower cumulative loss compared to Elo2k for every $0 < t < T$. For other datasets like Hearthstone, AlphaStar, 10,5-Blotto, go-dense, and mixedchess-dense, Elo2k achieves lower prediction errors than Elo at the final time $t = T$.

We can further investigate the results through the lens of regret minimization. We can see that the cumulative loss for each algorithm decreases over time, indicating the regret minimization effect of those algorithms. However the behavior for each algorithm at each sparsity level $t/N$ are not the same. These phenomenons are closely related to the sparsity level of dataset: when data is sparse, typically when $t/N < 1000$, the regret for Elo2k and Pairwise is so large, that even though the in hindsight baseline is much better, the cumulative loss $\frac{1}{t}\mathcal{L}_t$ for Elo2k will be large due to the large regret. Meanwhile, Elo achieves good performance due to its small regret. This may due to the fact that Elo, as online gradient descent for convex loss, has provable regret guarantees (Theorem 1) that ensures its performance. On the contrary, Elo2k suffer from its non-convex nature, and Pairwise has a much larger regret due to its parameter size of $N^2$ that is much larger then $N$, the Elo parameter size. When dataset is dense enough, for example, AlphaStar-dense, when $T/N > 1000$, we can see that the regret at time $T$ for both Elo2k and Elo are very small. In this regime, model capacity come into play. The baseline for Elo2k model is much smaller than the Elo counter part, therefore Elo2k shows better prediction accuracy than Elo. Among these dense datasets, LLM is special, since the Elo2k baseline and the Elo baseline are so close, that even the dataset is dense, Elo2k does not show any benefit.

We can futher see the influence of sparsity level, when we examine the dataset from Czarnecki et al. (2020): for AlphaStar, 5,4-Blotto and 10,5-Blotto, we create sparse version and dense version, where the underlying model is exactly the same, but "dense" version has 10 times sample size than "sparse" version. Through the comparison of these datasets, we can see that even under the same probabilistic model (which is non-BT), the behaviors of algorithms are still mainly affected by the sparsity level.

# F DETAILS OF SYNTHETIC EXPERIMENTS

## F.1 CONSTRUCTING $P$ FOR TRANSITIVE MODELS

We consider several ways of generating a SST/WST matrix $P$ w.r.t. the ordering $\pi(i) = N - i$. In the following constructions, we will firsts specify $P_{ij}$ for $i < j$, then make the matrix skew-symmetric by setting $P_{ij} = 1 - P_{ji}$ for $i > j$, and $P_{ii} = 0.5$.

**SST-byrow** We first generate a i.i.d. random sequence of length $N - 1$, each element is sampled from $\text{Uni}([0, 1])$. Then we sort this sequence in a descending order $r_1 \geq r_2 \geq \cdots \geq r_{N-1}$. We let $P_{ij} = 0.5 + 0.5 \times r_i$ for $i < j$.

**SST-bydiagonal** We first get the descending sequence $r_1 \geq r_2 \geq \cdots \geq r_{N-1}$ in the same way as SST-byrow. We let $P_{ij} = 0.5 + 0.5 \times r_{N-j+i}$ for $i < j$.

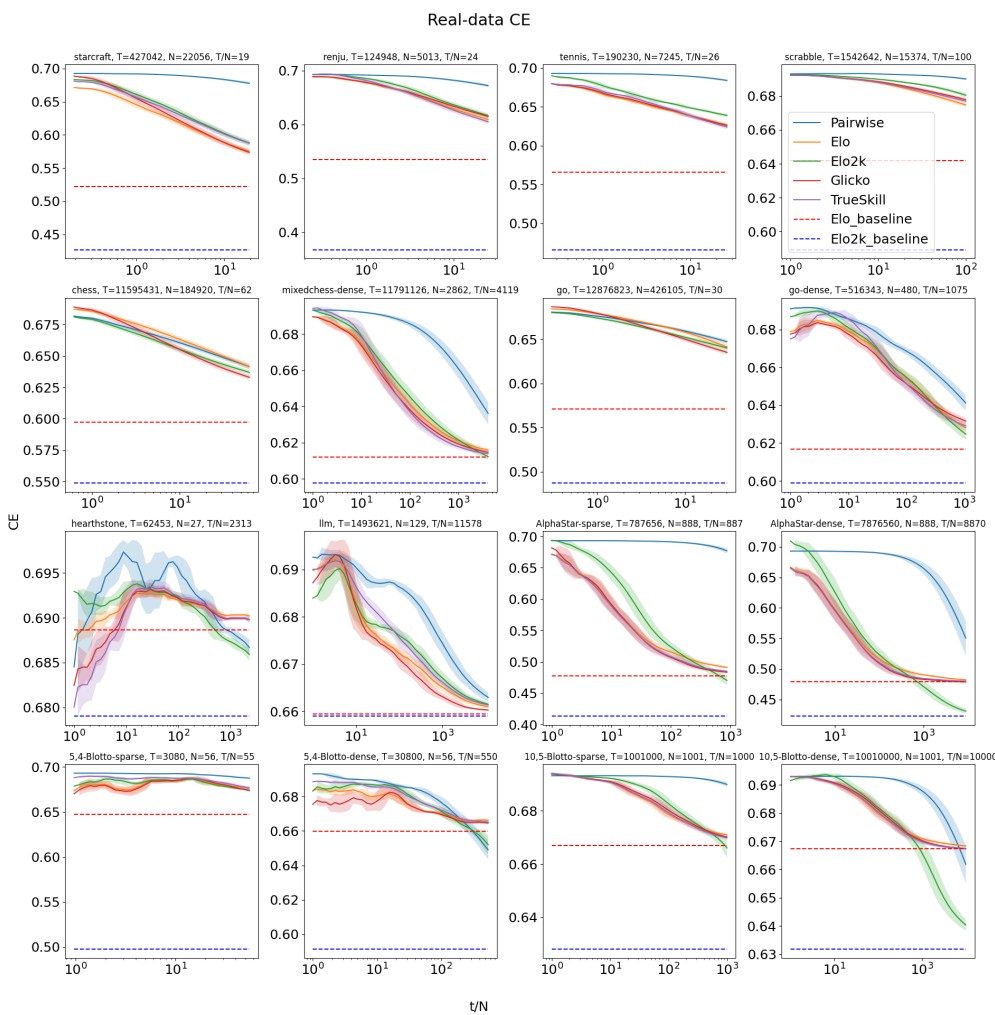

Figure 7: **In real datasets, sparsity strongly influences prediction.** Elo, TrueSkill, Glicko achieves the best prediction in sparse datasets, while Elo2k and Pairwise outperforms Elo and its variants in dense datasets.

**SST-byentry**   Following the "noisy sorting" model, we set $P_{ij} = 0.6$ for $i < j$.

**WST-byrow**   We first generate a i.i.d. random sequence of length $N - 1$, each element is sampled from $\mathrm{Uni}([0, 1])$. Then we sort this sequence in an ascending order $r_1 \le r_2 \le \cdots \le r_{N-1}$. We let $P_{ij} = 0.5 + 0.5 \times r_i$ for $i < j$.

**WST-bydiagonal**   We first get the ascending sequence $r_1 \le r_2 \le \cdots \le r_{N-1}$ in the same way as WST-byrow. We let $P_{ij} = 0.5 + 0.5 \times r_{N-j+i}$ for $i < j$.

**WST-byentry**   We set $P_{ij} = 0.5 + 0.5 \times U_{ij}$ for $i < j$, where $U_{ij} \sim \mathrm{Uni}[0, 1]$.

## F.2   CHOOSING THE BEST HYPERPARAMETER

For each algorithm (Elo, Elo2k, Glicko and TrueSkill), there are different hyperparameters that need to be chosen. We choose the parameters according to the follow criterion:

Let $CE_i := \frac{1}{t}\mathcal{L}_{t_i}$ be the CrossEntropy Loss at time $t_i$, where $\{t_i\}_{i=1}^n$ be the time steps we collect the loss. Define the threshold $M = \ln(2)$ (A purely random prediction will have a loss of $M$). The loss function $L(\mathbf{v})$ is given by: can be expressed as:

$$L = \sum_{i=1}^{30}\left(CE_i + 5(CE_i - M)\mathbb{I}(CE_i > M)\right).$$

We select the hyperparameter that minimizes the loss $L$. This loss function ensures that the chosen parameter achieves a consistently low average cross-entropy (CE) loss throughout the process while avoiding overfitting at some point (where $CE_i > M$ indicates overfitting).

### F.3 CREATING NON-STATIONARY DATASETS

Specifically, for modeling the varying player strength, for each type of underlying distribution (e.g., `SST`, `byrow`), we generate two matrices $P^0$ and $P^T$, and let $P^t = (1 - t/T) \times P^0 + (t/T) \times P^T$ be the win rate matrix at each time $t$. That is, $\mathbb{E}[o_t|i_t = i, j_t = j] = P_{ij}^t$.

For modeling non-trivial matchmaking, we construct the game dataset as the following: at each time point $t$, we sample $i_t \sim \text{Uni}([N])$, and then sample $j_t$ uniformly from the players that has ranking (by the real-time Elo score) within distance $K/2$ from $i_t$'s ranking. To be more concrete, let the ranking induced by Elo scores $(\theta[1], \cdots, \theta[N])$ be $\pi = (\pi(1), \cdots, \pi(N))$, a permutation of $(1, 2, \cdots, N)$ such that $\theta[\pi^{-1}(N)] > \theta[\pi^{-1}(N-1)] > \cdots > \theta[\pi^{-1}(1)]$. Then $j_t$ is chosen uniformly from the set $\{j \in [N] : |\pi(j) - \pi(i_t)| \leq K/2\}$. We choose $K = N/5$. After constructing such a game dataset, we fix this dataset and plot the performance of each algorithm

## G THEORY AND EXPERIMENTS FOR RANKING GIVEN BY ELO

### G.1 PROOFS FOR THEOREM 2

The formal version of Theorem 2 is stated as:

**Theorem G.1.** *Consider the population negative log-likelihood function of BT model $\mathbb{E}_q[\mathcal{L}(\theta)]$, where $q$ is the matchmaking distribution. Let*

$$\theta^* := \arg\min_{\theta \in \mathbb{R}^N} \mathbb{E}_q \mathcal{L}(\theta)$$

*be the population MLE. Then if $q$ is a product distribution, i.e., $q_{ij}$, the probability player $i$ plays with $j$, satisfies $q_{ij} = q_i q_j$ for any $i, j \in [N]$. Then the ranking given by $\theta^*$ is the same as the ranking given by the average win rate. This result hold for uniform matchmaking as a special case.*

*Proof.* With a slightly abuse of notation, we use $\theta_i$ to denote the $i-$th entry of $\theta$. Then

$$\mathbb{E}_q[\mathcal{L}(\theta)] = -\sum_{i,j \in [N]} q_{ij}(P_{ij}\ln(\sigma(\theta_i - \theta_j)) + P_{ji}\ln(1 - \sigma(\theta_i - \theta_j))).$$

Set it's gradient to zero, we have for each $i \in [N]$,

$$0 = \frac{\partial}{\partial \theta_i}\mathbb{E}_q[\mathcal{L}(\theta^*)] = \frac{\partial}{\partial \theta_i}(-\sum_{i,j \in [N]} q_{ij}(P_{ij}\ln(\sigma(\theta_i^* - \theta_j^*)) + P_{ji}\ln(1 - \sigma(\theta_i^* - \theta_j^*))))$$

$$= -\sum_{j \in [N]} q_{ij}(P_{ij} - \sigma(\theta_i^* - \theta_j^*)).$$

For the last equation we use the property that $\sigma'(t) = 1 - \sigma(t)$ and $P_{ji} = 1 - P_{ij}$. Since $q_{ij} = q_i q_j$, we can devide $q_i$ from both side and derive

$$\sum_{j \in [N]} q_j P_{ij} = \sum_{j \in [N]} q_j \sigma(\theta_i^* - \theta_j^*).$$

Notice that LHS $= \mathbb{E}_q \overline{P}[i]$ is the average win rate of player $i$ under matchmaking $q$. If we define

$$f(x) := \sum_{j \in [N]} q_j \sigma(x - \theta_j^*),$$

then $\mathbb{E}_q \overline{P}[i] = \text{LHS} = \text{RHS} = f(\theta_i^*)$. Notice that $f$ is a monotone increasing function, therefore the ranking given by $\mathbb{E}_q \overline{P}$ is the same as the ranking given by $\theta^*$. $\qquad\square$

## G.2 EXAMPLE WHERE ELO CONTRADICTS SST UNDER MATCHMAKING

**Example 1.** Consider the following $P$ matrix among 5 players that satisfies SST with $\pi(i) = 6 - i$.

$$P = \begin{bmatrix} 0.5 & 0.99 & 0.99 & 0.99 & 0.99 \\ 0.01 & 0.5 & 0.6 & 0.7 & 0.99 \\ 0.01 & 0.4 & 0.5 & 0.6 & 0.99 \\ 0.01 & 0.3 & 0.4 & 0.5 & 0.51 \\ 0.01 & 0.01 & 0.01 & 0.49 & 0.5 \end{bmatrix}.$$

Suppose that the matchmaking distribution is given by

$$Q = \begin{bmatrix} & & 0.125 & & \\ 0.125 & & & 0.125 & \\ & & & & 0.125 \\ & 0.125 & & & 0.125 \\ & & 0.125 & 0.125 & \end{bmatrix},$$

where the remaining entries are 0.

In this case, given infinite data,

$$\theta^{\mathrm{mle}} = [5.48, 0.89, 4.60, 0.04, 0],$$

which induces an inconsistent ranking $1 \succ 3 \succ 2 \succ 4 \succ 5$.

We also consider the regime where $T$ does not go to infinity. We conduct the following synthetic experiment: we generate random samples for $T = 10000$, following the $P$ and $Q$ in Example 1. Then we construct confidence interval for each player's Elo score by bootstrapping (following the procedure in chatbot arena): we sample $T = 10000$ times with replacement from the original created random samples. We create 100 such bootstrap samples. For each of these samples, we can compute an Elo score (we regularize the scores so that player 5 always has score 0). Then for each player's Elo score in 100 different samples, we can compute the 0.05 quantile and 0.95 quantile for these scores, therefore give a confidence interval for each player's score. The resulting confidence interval for each player is: $[4.86, 5.33], [0.72, 1.17], [4.18, 4.68], [-0.07, 0.31], [0.00, 0.00]$. From these confidence intervals, we can confidently say the Elo scores give the ranking $1 \succ 3 \succ 2 \succ 4$, which is inconsistent for players $\{1, 2, 3, 4\}$.

