# OpenReview forum: "Is ELO Rating Reliable? A Study Under Model Misspecification"
_ICLR.cc/2026/Conference — Submitted to ICLR 2026_

### Official Review · Reviewer_erau · 2025-10-28

**Soundness:** 3
**Presentation:** 3
**Contribution:** 3
**Rating:** 6
**Confidence:** 4

**Summary:**

This paper presents a thorough empirical and theoretical investigation into the performance and reliability of the Elo rating system in real-world scenarios where its foundational assumptions—specifically, the Bradley-Terry (BT) model and stationarity of player skills—are violated. The authors first demonstrate, via a likelihood ratio test, that multiple real-world game datasets (Chess, Go, LLM Arena, etc.) significantly deviate from the BT model and exhibit non-stationarity. The central, counter-intuitive finding is that despite this model misspecification, the simple Elo algorithm (and its close variants like Glicko and TrueSkill) often outperforms more complex and expressive models like multi-dimensional Elo (Elo2k) and pairwise win-rate models in terms of win-prediction accuracy, particularly in common "sparse" data regimes. The work concludes that Elo's empirical success is not due to its fidelity to the BT model but to its robust online learning properties, which are particularly effective in the sparse data environments typical of real-world applications.

**Strengths:**

1. The core result—that a misspecified, simple model (Elo) outperforms well-specified, complex models on the very non-BT data they were designed for—is surprising, important, and challenges the conventional understanding of Elo.

2. The paper does not just present an empirical anomaly but provides a deep and convincing explanation by connecting Elo to online learning theory (regret minimization) and systematically analyzing the critical role of data sparsity. This three-pronged approach (theory, synthetic exps, real-world exps) is very effective.

3. The study is exceptionally thorough. The use of a likelihood ratio test to formally reject the BT model across many datasets is methodologically sound. The synthetic experiments are well-designed to isolate factors like transitivity type (SST/WST) and sparsity. The consideration of different matchmaking schemes and non-stationarity adds significant practical relevance.

4. The paper is generally well-written and structured. The narrative flows logically from the empirical puzzle to its resolution. The figures effectively support the main claims about sparsity and regret.

5. The findings have immediate practical implications for practitioners using rating systems in games, sports, or LLM evaluation (like Chatbot Arena). The paper provides clear guidance on when to prefer simple models like Elo over more complex alternatives.

**Weaknesses:**

1. While the non-convexity of Elo2k is mentioned as a reason for its higher regret, the analysis could be deeper. Is the issue solely the number of parameters, or is the optimization landscape inherently problematic, especially early in learning? A brief discussion or citation on the challenges of training low-rank models in online, sparse settings would strengthen this point.

2. The paper uses the normalized time t/N (total games per player) as a proxy for sparsity. However, the effective sparsity is also determined by the matchmaking distribution. A highly non-uniform matchmaking (e.g., players only facing a small subset of others) could create a sparse environment even if t/N is large. The analysis could more explicitly discuss this interaction.

3. The "Accuracy" column reports cross-entropy loss (lower is better), but the label says "Accuracy." This should be corrected for clarity. Furthermore, for large datasets like Chess and Go, the Pairwise results are missing ("-"), likely due to computational infeasibility. This should be explicitly stated in the caption or a footnote.

**Questions:**

1. The paper mentions selecting the best hyperparameters for each algorithm. Given that the performance of online learning algorithms can be sensitive to learning rates, to what extent does Elo's superiority depend on careful tuning? Did the more complex models (Elo2k) require more sensitive tuning, and could this be a hidden factor in their poorer performance in sparse regimes?

2. Elo's OGD interpretation relies on the convexity of the logistic loss. Have the authors considered or experimented with other link functions? Is part of Elo's robustness attributable to the properties of the logistic function itself, independent of the BT model assumption?

3. The comparison is made against Elo2k and the full Pairwise model. Were other, potentially more robust, regularized versions of complex models (e.g., with strong L2 regularization or Bayesian priors for Elo2k) explored? Could such methods bridge the performance gap in sparse data?

---

### Official Review · Reviewer_SLLb · 2025-10-30

**Soundness:** 1
**Presentation:** 2
**Contribution:** 1
**Rating:** 4
**Confidence:** 4

**Summary:**

This paper investigates the reliability of the Elo rating system under model misspecification and non-stationarity. The authors empirically demonstrate that real-world game datasets (e.g., Chess, Go, LLM Arena) significantly deviate from the Bradley-Terry (BT) model and exhibit non-stationary player skills and matchmaking. Surprisingly, despite these deviations, Elo outperforms more complex models (e.g., mElo, Pairwise) in win-rate prediction.

**Strengths:**

This paper's main claim is that the Elo is still reliable in misspecified and non-stationary settings. However, this work doesn't clearly define what the misspecified and non-stationary indicate. I think formally displaying the definition of misspecified and stationary/non-stationary is important.

Synthetic experiments and real-world results demonstrate Elo’s robustness in sparse regimes

**Weaknesses:**

The misspecification to BT mode is not a surprising finding. This paper says Elo's loss function is convex, but Elo2k isn't convex, therefore the Figure 1 gets Elo outperforms Elo2k in sparse datasets. But we take more care of the discussion of the Elo and or other baselines. So if the Theorem 1. is just used to claim the superiority of Elo to Elo2k, this theorem isn't helpful too much and isn't very novel.

[1] Ding, Q.; Hsieh, C.-J.; and Sharpnack, J. 2021. An efficient algorithm for generalized linear bandit: Online stochastic gradient descent and thompson sampling. In International Conference on Artificial Intelligence and Statistics (AIS- TATS), 1585–1593.

The theoretical analyses seem right and solid, but the technique novelty seems lacking. [1] has given a regret bound under the convex and gradient descent setting. [1] Ding, Q.; Hsieh, C.-J.; and Sharpnack, J. 2021. An efficient algorithm for generalized linear bandit: Online stochastic gradient descent and thompson sampling. In International Conference on Artificial Intelligence and Statistics (AIS- TATS), 1585–1593.

This paper shows that the Elo gains very similar performance with other baselines including Trueskill Elo2k Glicko, which can't support the claim "Elo’s superior performance in prediction compared to more complex rating systems

The BT model is a very strict and ideal assumption, and most real-world datasets can't fit this model, thus the misspecification problem isn't an interesting finding for me.

The results in tab 2 show very similar performance between baselines, I can't much agree with the claim that "Elo and “Elo-like” ratings outperform more complexity rating systems such as Elo2k and Pairwise". The Pairwise baselines just statistic the empirical winning rates between limited players, and exhibit superior performance than other baselines, so this indicates the carefully designed rating system like trueskill is meaningless right?

**Questions:**

I'm sorry that I haven't found a significant improvement compared to the previous draft. And this paper doesn't include the conclusion section.

**Details Of Ethics Concerns:**

Don't include LLM usage statement, this disobeys the ICLR rules

---

### Official Review · Reviewer_31cV · 2025-11-01

**Soundness:** 2
**Presentation:** 1
**Contribution:** 1
**Rating:** 2
**Confidence:** 3

**Summary:**

The paper argues that the Elo rating system, viewed as a maximum-likelihood estimator under the Bradley–Terry (BT) model, is theoretically flawed since BT is systematically rejected by real data. It reinterprets Elo as an online learning algorithm equivalent to online gradient descent on the logistic loss, showing it has no-regret guarantees. Empirically, the authors test BT realizability using a two-dimensional likelihood-ratio test and show widespread rejection, yet Elo performs comparably to more complex variants (Elo2k, mElo). Theoretical results demonstrate asymptotic ranking consistency under uniform matchmaking, but possible failure otherwise.

**Strengths:**

# originality

Introduces a likelihood-ratio test for BT realizability in sparse or adaptive data.

# quality

Provides a ranking theorem.

The empirical study covers several real datasets and synthetic settings.

# clarity
Proofs and definitions are mostly self-contained.

# significance

Relevant to practical ranking systems in games and competitions.

**Weaknesses:**

# originality

The interpretation of Elo as an online convex optimization algorithm that performs gradient descent on logistic loss is already well-known (e.g., blog post by Steven Morse, 2018; standard OGD literature). The paper restates this equivalence and applies a textbook regret bound without clear acknowledgment of prior art. Hence, the theoretical novelty is limited to presentation rather than substance.

# quality

Overstated empirical claims: Table 1 shows that Elo outperforms Elo2k in only 5 of 8 datasets, and the differences are confined to the third decimal place of cross-entropy (around $10^{-3}$). No confidence intervals or significance tests are reported, so these tiny margins may be noise.

The paper’s language (“consistently better,” “often outperforms”) exaggerates the empirical evidence. The accurate interpretation is that Elo performs on par with Elo2k despite its simplicity.

The paper does not provide formal goodness-of-fit tests for the more complex alternatives (Elo2k, Pairwise), limiting the strength of conclusions about their supposed misspecification.

The choice of two-dimensional augmentation for the LRT is explained but somewhat arbitrary. Power and sensitivity analyses for higher-dimensional variants are not explored.

Empirical sections could report hyperparameter sensitivity and learning-rate robustness, since these affect regret and performance.

# clarity

The distinction between “Elo not fitting BT” and “Elo good in prediction” could be made earlier and more explicitly to avoid confusion.


# significance

The conclusion that “Elo cannot be blindly trusted” is correct, but too generic -- it applies to any modeling effort. The real contribution is identifying precise conditions (matchmaking bias, SST violation) under which it fails.

The paper would gain practical impact by providing quantitative heuristics (e.g., thresholds on T/N or per-pair coverage) to help practitioners decide when a dataset is “sparse enough” for Elo’s simplicity to be advantageous.

**Questions:**

See weaknesses

---

### Official Review · Reviewer_NeqQ · 2025-11-07

**Soundness:** 3
**Presentation:** 3
**Contribution:** 3
**Rating:** 6
**Confidence:** 3

**Summary:**

This paper demonstrates theoretically and empirically why the Elo rating performs better than more complex models even though, unlike in their case, many real problems violate Elo’s underlying modelling assumptions (the Bradley-Terry model). The authors show this by reformulating Elo as a regret minimising online gradient descent procedure. They also demonstrate that prediction and ranking accuracy are correlated.

The method is well motivated, clear and convincing, though the topic is a little niche and it would be good to add a section clarifying its impact (perhaps mentioning timeliness due to LLMs). There are some other relatively small additions that could further improve it.

**Strengths:**

* The paper is well written and clear
*  The problem (within the topic of Elo) is well motivated and the method/story is presented very intuitively
*  Rigorous hypothesis testing helps to motivate the problem by showing the violation of BT
*  The main theoretical contribution (regret-based Elo) is simple and elegantly bridges probabilistic and optimisation views of rating
*  The theory is well supported by the empirical results
*  Several rating algorithms are compared on a few datasets

**Weaknesses:**

* The paper ends abruptly without a conclusion due to hitting the page limit (also no reproducibility statement)
* Missing discussion of the impact of this paper and why it is important (Elo has been trusted for a long time, why do we need to study it now?)
* In equation 3, the model misspecification error is not defined
* The choice of datasets are a bit arbitrary. It would be good to explain analytically why they violate the BT conditions and compare them to games that don’t, for better support of your claim that Elo performs well due to sparsity (rather than stationarity)
* It would be good to have more LLM-based datasets since the timeliness of this paper is possibly due to Elo’s use for evaluating LLMs
* The paper needs a proofread to correct some small grammatical errors (e.g. missing ‘the’s and to say ‘preliminaries’ instead of ‘preliminary’)

**Questions:**

See questions in weakness.

---

### Meta-Review · Area_Chair_vtYF · 2026-01-08

**Summary:**

This paper provides both theoretical and empirical evidence for why the Elo rating system outperforms more complex models, despite the fact that many real-world scenarios violate the fundamental modeling assumptions of Elo, which is based on the Bradley-Terry model. The authors achieve this by reformulating the Elo system as a regret-minimizing online gradient descent procedure. They also establish a correlation between prediction accuracy and ranking accuracy.The methodology is well-founded, clearly presented, and persuasive. However, the topic may be somewhat specialized, and it would be beneficial to include a section that clarifies its broader impact, possibly emphasizing its relevance in light of the rise of large language models (LLMs). Additionally, there are a few minor enhancements that could further refine the paper.

**Reviewer Concerns:**

The paper concludes abruptly without a summary because it reaches the page limit, and it also lacks a reproducibility statement. There is a notable absence of discussion regarding the significance of this research and its relevance; given that the Elo rating system has been widely trusted for many years, it raises the question of why it is necessary to study it at this time.Additionally, the model misspecification error in Equation 3 is not defined. The selection of datasets appears somewhat arbitrary; a detailed analytical explanation of how they violate the Bradley-Terry (BT) conditions would strengthen the argument. Comparing these datasets to games that do not violate these conditions would further substantiate the claim that Elo's effectiveness is attributed to sparsity rather than stationarity.In light of the paper's relevance, incorporating more datasets based on large language models (LLMs) would be beneficial, as the current timeliness of the study may be linked to Elo's application in evaluating LLMs. Finally, the paper requires proofreading to address minor grammatical issues, such as missing articles and using the term "preliminaries" instead of "preliminary."

**Reviewer Scores:**

The scores of the reviewers are 6,2,4,6, and the authors do not post rebuttal.

---

### Decision · Program_Chairs · 2026-01-26

Reject